# RNA Polymerase III-Transcribed RNAs in Health and Disease: Mechanisms, Dysfunction, and Future Directions

**DOI:** 10.3390/ijms26125852

**Published:** 2025-06-18

**Authors:** Longjie Sun, Mingyue Chen, Xin Wang

**Affiliations:** 1Key Laboratory of Systems Health Science of Zhejiang Province, School of Life Science, Hangzhou Institute for Advanced Study, University of Chinese Academy of Sciences, Hangzhou 310024, China; sljc000@ucas.ac.cn; 2Sanya Institute of China Agricultural University, Sanya 572025, China; mychenlj@163.com

**Keywords:** RNA polymerase III, transcriptome, non-coding RNAs, tsRNAs, disease

## Abstract

RNA polymerase III (Pol III) transcribes a broad spectrum of non-coding RNAs, including transfer RNAs (tRNAs), 5S ribosomal RNA (5S rRNA), U6 small nuclear RNA (U6 snRNA), and a range of regulatory RNAs (7SK, 7SL, RMRP, RPPH1, Y RNA, vault RNA, Alu, BC200, snaR, and nc886). These RNAs are integral to fundamental cellular processes, including transcription and translation, RNA processing and stability, and cytoplasmic protein targeting. Among them, tRNA-derived small RNAs (tsRNAs) have recently emerged as critical regulators across a wide array of biological contexts. Increasing evidence links the dysfunction of Pol III transcripts to human diseases, particularly genetic disorders and cancer. In this review, we provide a comprehensive overview of Pol III-transcribed RNAs, their biogenesis and regulatory mechanisms, and their biological functions. We also explore emerging insights into the disease relevance of Pol III-transcribed RNAs and discuss their potential implications for future research and therapeutic development.

## 1. Introduction

RNA polymerase III (Pol III) is a pivotal yet historically underappreciated component of the eukaryotic transcriptional machinery. Distinct from its counterparts Pol I and II—which are primarily responsible for the synthesis of ribosomal RNAs (rRNAs) and messenger RNAs (mRNAs), respectively—Pol III is dedicated to transcribing a diverse repertoire of small non-coding RNAs (ncRNAs) that are essential for numerous cellular processes [1,2]. These include canonical molecules such as tRNAs [3], 5S rRNA [4], and U6 snRNA [5], as well as regulatory and context-specific RNAs including 7SK, 7SL, RMRP, RPPH1, Y RNA, vault RNA, Alu, BC200, snaR, and nc886 [6,7,8,9,10,11,12,13,14,15,16]. Despite their relatively small size, these RNAs orchestrate fundamental functions in gene regulation, RNA maturation and stability, protein synthesis, and other fundamental biological processes [17].

The functional significance of Pol III is underscored by its most extensively studied products: tRNAs and 5S rRNA [18,19]. tRNAs are the essential adapter molecules mediating the translation of messenger RNAs in the decoding center of the ribosome [20]. Recent advances have characterized the spatiotemporal expression dynamics and post-transcriptional modifications of tRNAs during development [21,22]. Meanwhile, 5S rRNA, distinct from other rRNAs transcribed by Pol I, plays dual functions—both as a structural component of the large ribosomal subunit and as a facilitator of ribosome biogenesis by contributing to peptidyl transferase center assembly [23]. As research progresses, an increasing number of Pol III-derived RNAs have been discovered and shown to participate in diverse biological processes [24,25]. It is widely known that the transcription of Pol III-dependent RNAs is tightly regulated, and any disruption in this process can lead to severe consequences [26]. Over the past few decades, research has revealed that dysregulation of Pol III transcription is associated with a variety of diseases, including developmental disorders, diseases, and cancer [17,19,27,28,29].

In addition, the evolving understanding of Pol III-transcribed RNAs is reshaping conventional views of RNA biology. tRNAs, once regarded as passive adapters in protein synthesis, are now recognized as precursors to a class of regulatory molecules, tsRNAs, that emerge through stress-induced cleavage by enzymes such as angiogenin [30]. These tsRNAs exert diverse functions, including global translational repression and retrotransposon silencing [31]. The discoveries of tsRNAs underscore a paradigm shift: Pol III products are not mere executors of cellular routines but active participants in signaling networks, blurring the lines between coding and non-coding RNA biology.

This review provides a comprehensive overview of the Pol III transcriptome and its roles in cellular homeostasis and disease. We focus on their transcriptional regulation, context-dependent functions, and mechanistic links to pathologies such as cancer and neurodevelopmental disorders. By synthesizing current evidence and highlighting critical knowledge gaps, this work aims to inform future investigations into diagnostic and therapeutic strategies targeting Pol III-associated ncRNAs.

## 2. RNAs Transcribed by Pol III

Since the discovery of tRNAs in 1958, an expanding repertoire of non-coding RNAs transcribed by Pol III has been identified (Figure 1). The transcription of RNA Pol III-dependent RNAs is a tightly regulated process encompassing three key stages: initiation, elongation, and termination. During initiation, pre-initiation complexes (PICs) assemble at promoter regions via mechanisms that differ by promoter type. Type 1 promoters (e.g., 5S rRNA genes) contain internal control regions (ICRs) where TFIIIA binds, enabling recruitment of TFIIIC and subsequently TFIIIB—a complex composed of BRF1, TBP (TATA-binding protein), and BDP1—which positions Pol III at the transcription start site. Type 2 promoters (e.g., tRNA genes) also contain ICRs but lack TFIIIA involvement; instead, TFIIIC directly recruits TFIIIB. In contrast, type 3 promoters (e.g., U6 snRNA genes) employ upstream sequence elements (USEs) and TATA boxes recognized by the SNAPc complex, which directly recruits TFIIIB and Pol III. Following PIC assembly, Pol III initiates RNA synthesis, with elongation driven by its catalytic core. Termination is triggered by a thymidine-rich sequence (T-stretch) in the DNA template, leading to RNA release with a characteristic 3′ poly-U tract. This multi-layered regulation ensures precise control over Pol III activity across diverse RNA classes. Pol III transcribes a wide variety of non-coding RNAs, including tRNAs, 5S rRNA, U6 snRNA, 7SK snRNA, 7SL RNA, RMRP, RPPH1, Y RNA, vault RNA, Alu, BC200, snaR, and nc886, some of which can be cleaved into small fragments that participate in a wide range of cytological processes. In particular, tRNA-derived fragments are the most representative and are classified into five categories: 5′-tRFs, 3′-tRFs, 5′-halves, 3′-halves, and i′-tRFs [32,33]. In addition, vault RNAs derive fragments that are svRNA1, svRNA2, svRNA3, and svRNA4, while Y RNAs derive fragments that are 3′ysRNAs and 5′ysRNAs [34,35]. The major classes of Pol III-transcribed RNAs and their derived fragments are summarized below (Figure 2).

### 2.1. tRNAs and tsRNAs

tRNAs are indispensable adaptors in the translation apparatus, delivering amino acids to the ribosome for polypeptide synthesis [3,36]. The biosynthesis of tRNAs is precisely regulated by RNA Pol III, which initiates transcription at highly conserved intragenic sequence elements within tRNA genes, specifically the A-box (TGGCNNAGTGG) and B-box (GGTTCGANNCC) [37]. Pol III transcribes all nuclear-encoded tRNAs, which are characterized by their cloverleaf secondary structure and conserved sequence motifs [17]. In 2009, Lee et al. identified a previously unrecognized class of small RNAs (tsRNAs) that rank second only to microRNAs (miRNAs) in cellular abundance [38]. Once categorized as transient byproducts of tRNA catabolism, tsRNAs are now recognized as evolutionarily conserved regulators of diverse biological pathways [39]. tsRNAs are a new type of ncRNAs produced by the specific cleavage of precursor or mature tRNAs [40]. Mature tRNAs can generate five subtypes of tsRNAs: 5′-half, 3′-half, 5′-tRF, 3′-tRF, and i-tRF [32,33]. Apart from the standard classification of tRFs, there are atypical classes of tRFs generated from pre-tRNAs, such as tRF-1 [38]. Advances in high-throughput sequencing have significantly expanded the catalog of functionally characterized tsRNAs, revealing their critical roles in diverse physiological and pathological processes [41]. Part of the tsRNA binds to PIWIL2 and acts as a signaling molecule for cytokines in the form of piRNA to modulate the expression of vital membrane with lipid Ag-presenting proteins [42]. In summary, tRNAs and tsRNAs modulate essential cellular activities, such as proliferation, differentiation, and homeostasis, through multifaceted molecular mechanisms, thereby establishing their significance as pivotal regulators in diseases, including cancer [20,43,44,45,46].

### 2.2. 5S rRNA

In eukaryotic genomes, 5S rDNA exists as tandemly repeated arrays, forming distinct chromosomal clusters [47]. The transcription of 5S rRNA requires the coordinated activity of two conserved promoter elements: the A-box and a type 1-specific C-box [48]. The latter directly mediates the recruitment of TFIIIA, a core transcription factor essential for the assembly of the transcriptional pre-initiation complex and for initiating Pol III-dependent transcription [49]. The mature 5S rRNA molecule, approximately 120 nucleotides in length with a molecular weight of ~40 kDa, displays a conserved secondary structure organized into five double-helical domains (designated I–V) interconnected by four loop regions [23]. These loops are classified into two hairpin-type and two internal loop-type configurations, with loop A functioning as a pivotal junction bridging stems I, II, and V [50]. Functionally, 5S rRNA is a component of the large ribosomal subunit and is essential for ribosome assembly and function [23]. It forms a complex with ribosomal proteins and other rRNAs to facilitate protein synthesis.

### 2.3. U6 snRNA

U6 snRNA is a key component of the spliceosome and was originally discovered by co-immunoprecipitation with other uridine-rich snRNAs [5]. In humans, U6 snRNA is encoded by multiple genes with differential transcriptional activities, predominantly regulated by type 3 promoters [51]. Its maturation involves two key steps: 5′ monomethyl capping mediated by MePCE, and 3′ terminal processing via sequential oligouridylation by TUT1 (a uridylyl transferase) followed by exonuclease trimming via USB1 [52,53,54]. Beyond these maturation steps, U6 snRNA undergoes extensive chemical modifications that are critical for its function within the spliceosome. These include pseudouridylation, 2′-O-methylation, and site-specific m^6^A/m^2^G methylation [55]. Multiple studies demonstrate that LARP7 participates in U6 snRNA 2′-O-methylation, which regulates diverse biological processes [56,57,58]. Similarly, THUMPD2 catalyzes m^2^G methylation within the catalytic core of U6 snRNA, modulating spliceosome function and influencing processes such as retinal development and degeneration [59]. METTL16, another RNA methyltransferase, installs m^6^A marks on U6 snRNA and regulates the intron retention of S-adenosylmethionine (SAM) biosynthesis [60]. Structurally, U6 comprises ~100 nucleotides, while the minor spliceosome-specific U6atac variant spans ~120 nucleotides with an extended 3′ tail and an auxiliary stem–loop essential for minor spliceosome assembly [61]. Notably, U6atac’s 5′ monomethyl cap recruits unique stabilizing factors (RBM48 and ARMC7) critical for maintaining the catalytic core conformation of the minor spliceosome [62]. Together, U6 and U6atac snRNAs, both transcribed by Pol III, serve as core catalytic components of the major and minor spliceosomes, respectively [17].

### 2.4. 7SK snRNA

Pol III mediates the tightly regulated synthesis of 7SK snRNA, a highly structured, non-coding RNA approximately 332 nucleotides in length. Unlike most Pol III transcripts that participate in RNA processing or translation, 7SK snRNA exerts pivotal regulatory functions in Pol II transcription [63]. As a central coordinator of transcriptional elongation, 7SK inhibits positive transcription elongation factor b (P-TEFb) to suppress Pol II activity and forms dynamic RNP complexes with core components MePCE and LARP7 [64,65]. Notably, HIV-1 exploits this regulatory axis by hijacking 7SK RNP complexes to enhance viral replication [66]. In response, LARP7, the core component of 7SK snRNP, was able to inhibit HIV-1 replication by liquid–liquid phase separation [67]. In addition, the 7SK snRNP interacts with the Survival Motor Neuron (SMN) complex, which mediates spliceosomal snRNP biogenesis, to regulate snRNP production [68]. These findings position 7SK as a multifaceted regulator of gene expression, linking Pol III transcription with Pol II elongation control and broader RNA maturation processes.

### 2.5. 7SL RNA and Its Evolutionary Derivatives (Alu, BC200, and snaR RNAs)

7SL RNA, the RNA component of the signal recognition particle (SRP), constitutes the structural and functional core of the SRP RNP complex, which mediates the recognition and endoplasmic reticulum (ER)-targeting of nascent secretory and membrane proteins [17]. Human 7SL RNA genes employ type 2 hybrid promoters, combining intragenic and upstream regulatory elements [69]. These loci encode precursor transcripts that mature through 3′-oligo(U) trimming and adenine addition, yielding functional RNAs of ~300 nucleotides [70]. Importantly, 7SL RNA served as the evolutionary template for several primate-specific, non-coding RNAs, most notably Alu elements, BC200, and snaR RNAs. Alu elements, derived from dimeric 7SL RNA sequences, comprise the most abundant short interspersed nuclear elements (SINEs) in the human genome, with approximately 1 million copies accounting for ~11% of genomic DNA [71,72]. The Alu elements were a ~280-nucleotide DNA repeat with A-box and B-box type 2 promoter elements [72]. Using the Alu element transcribed by Pol III, the resulting Alu ncRNA can continue to regulate transcription and translation. Specifically, Alu RNA directly binds two Pol II molecules through distinct interaction interfaces mediated by its left (Alu-LA, also known as scAlu) and right (Alu-RA) arm subdomains [73]. Although both monomeric subdomains of Alu RNA adopt identical elongation complex (EC)-like conformations when bound to RNA polymerase II (Pol II), their transcriptional repression function depends on differential interactions with distinct TFIIF subunits, likely facilitated by RNA-binding activity within TFIIF domains [74]. Moreover, Alu-derived sequences have been co-opted into novel non-coding RNA genes through genomic exaptation [17]. For example, BC200 RNA, transcribed from the BCYRN1 locus, evolved from an Alu monomer through sequential nucleotide substitutions and insertions, retaining sequence homology with 7SL and Alu while acquiring divergent 3′-terminal structural features [75]. Like its progenitor, BC200 is transcribed by Pol III from a type 2 hybrid promoter composed of both intragenic and upstream elements [13,76]. A novel, stable, truncated form of BC200 exists at only 200 nt, and this RNA fragment is referred to as BC120. BC120 is expressed in a variety of normal human tissues and is also elevated in ovarian cancer. BC120 is highly stable and evades knockdown strategies targeting the 3′ unique sequence of BC200, suggesting a distinct functional role [77]. Another evolutionary product of Alu sequence remodeling is small NF90-associated RNA (snaR), a ~120-nt Pol III-transcribed RNA that originated through iterative deletions and expansions of a monomeric Alu sequence [14]. snaR was initially identified as a primate-specific ncRNA upregulated in Epstein–Barr virus (EBV)-infected B cells [78,79]. Its transcription depends on type 2 promoters featuring intragenic A-box and B-box elements [17]. Evolutionarily restricted to higher primates, similar to BC200 RNA, snaR exhibits tissue-selective expression patterns, with the specific isoform snaR-A showing robust expression in testis and aberrant emergence in human cancers [80]. Collectively, 7SL RNA and its evolutionary derivatives—Alu, BC200, and snaR—illustrate the functional plasticity of Pol III-derived transcripts. These RNAs, once regarded as structural or parasitic, have been recurrently co-opted into host regulatory networks, particularly in translational control and cancer biology [17,27].

### 2.6. RPPH1 and RMRP RNAs

Pol III-transcribed RPPH1 (also known as H1) ncRNA functions as the catalytic core of RNase P, a site-specific endonuclease essential for processing pre-tRNA 5′-leader sequences [81]. This 341-nucleotide RNA is transcribed from a single locus harboring type 3 upstream promoter elements and adopts a conserved bipartite structure with two functional domains: the specificity (S) domain responsible for substrate recognition and the catalytic (C) domain containing the tRNA cleavage active site [82]. RPPH1 associates with multiple protein subunits, including RPP14, RPP20, RPP21, RPP25, RPP29, RPP30, RPP38, RPP40, hPOP1, and hPOP5, which collectively stabilize its conformation and enhance pre-tRNA binding and processing [83]. Beyond its canonical RNase P role, RPPH1 interacts with distinct proteins and miRNAs to regulate cellular proliferation and growth through mechanisms that remain incompletely characterized [84,85,86,87].

The eukaryotic ribozyme RNase MRP shares functional parallels with RNase P, utilizing Pol III-transcribed RMRP ncRNA to mediate pre-rRNA cleavage [8]. Originally identified as nucleolar RNA 7–2 in ribonucleoprotein (RNP) immunoprecipitates, RMRP (~270 nucleotides) resembles RPPH1 in its type 3 promoter architecture and secondary structure, including a conserved catalytic C-domain with extended stem–loop motifs [17]. However, RMRP diverges from RPPH1 in its specificity (S) domain, which remodels substrate-binding pockets to confer rRNA selectivity [88]. In addition, nuclear-expressed RMRP promotes its nuclear export and mitochondrial localization through HuR and GRSF1, inhibiting oxygen consumption rates and mitochondrial DNA replication priming [89]. While RMRP’s post-transcriptional processing remains poorly characterized, m^6^A methylation has been shown to stabilize its mature form, suggesting epitranscriptional regulation of RNase MRP activity [90]. Together, RPPH1 and RMRP exemplify how Pol III-derived catalytic RNAs integrate RNA metabolism with broader cellular regulatory networks. By participating in essential processing steps for tRNAs and rRNAs—and potentially interacting with key signaling pathways—these non-coding RNAs function as critical nodes connecting Pol III transcription to cell growth, differentiation, and disease.

### 2.7. Y RNA and Y RNA-Derived Small RNAs (ysRNAs)

Y RNAs are a conserved class of Pol III-transcribed non-coding RNAs originally identified as core components of Ro60-containing ribonucleoprotein (RoRNP) complexes through their reactivity with autoantibodies from patients with systemic lupus erythematosus [11,17]. In humans, four Y RNA variants (84–112 nucleotides) are transcribed from single-copy loci harboring type 3 promoters [91]. Beyond their canonical RNP assembly functions, Y RNAs are increasingly recognized as multifunctional regulators of cellular homeostasis. Several studies have implicated them in modulating the fidelity of DNA replication, as well as in ensuring proper RNA processing under both basal and stress conditions—functions that are tightly linked to tumorigenesis and cellular stress responses [92]. Under cellular stress, Y RNAs undergo site-specific cleavage at their 3′ and 5′ termini to generate ysRNAs [93]. These fragments, initially dismissed as degradation byproducts, are now recognized as critical regulators of apoptosis and related biological pathways [94]. Mechanistically, 3′-end ysRNA biogenesis in *RNY5* (transcribed Y RNA) requires an internal loop adjacent to a conserved 5–6 nucleotide stem (S2 domain), while 5′-end processing depends on a UGGGU motif spanning positions 22–25 [35]. Notably, these 5′-derived ysRNAs exhibit potent apoptotic activity [93]. Both 3′- and 5′-terminal *RNY5*-derived ysRNAs depend on RO60 binding for their biogenesis, whereas ribonuclease L (RNASEL) displays species-specific roles: dispensable for human 3′-end cleavage but contributory to murine Y RNA processing [35]. Thus, Y RNAs and ysRNAs functionally regulate diverse cellular processes such as DNA replication, RNA homeostasis, and stress adaptation mechanisms.

### 2.8. Vault RNA (vtRNA) and vtRNA-Associated Small RNAs (vtsRNAs)

vtRNAs, initially identified as structural components of vault ribonucleoprotein (RNP) complexes, are transcribed from three tandemly arranged Pol III-dependent loci (*VTRNA1-1*, *VTRNA1-2*, *VTRNA1-3*) on human chromosome 5 [95]. These genes exhibit type 2 hybrid promoter architectures integrating intragenic A/B-box motifs and upstream regulatory elements [96]. Among them, *VTRNA1-1* exhibits the highest expression, potentially due to a unique downstream B2-box element that enhances transcriptional efficiency [17]. Mature vtRNAs (88–100 nucleotides) adopt conserved stem–loop secondary structures with a central variable loop, stabilized by NSUN2-mediated m^5^C methylation—a modification critical for generating vtRNA-derived small RNAs (vtsRNAs) [34,97]. Processing of vtRNAs into vtsRNAs is tightly regulated. For instance, the spliceosomal protein SRSF2 binds to vtRNA1-1 and suppresses its cleavage into vtsRNAs, thereby modulating the abundance of these regulatory fragments. Notably, certain vtsRNAs mimic the function of small nucleolar- or miRNA-like RNAs, such as miR-snaR, suggesting an expanded repertoire of RNA-mediated regulatory activities [34,98]. In mice, only a single vtRNA homolog, *Vaultrc5*, has been identified [97]. Interestingly, *Vaultrc5* knockout mice are viable and display no overt developmental abnormalities, although they exhibit a modest reduction in platelet counts, suggesting a non-essential yet potentially modulatory role in hematopoiesis [99]. To date, studies have identified vault RNAs (vtRNAs) as critical regulators of antiviral, tumorigenesis, apoptosis, therapy resistance, and autophagy, positioning them as promising therapeutic targets for oncology [17,100].

### 2.9. nc886

nc886 is a 101-nucleotide non-coding RNA transcribed by RNA polymerase III under the regulation of a type 2 hybrid promoter architecture [101]. Also known as vault RNA2-1 or pre-miR-886, nc886 presents a taxonomic ambiguity in current RNA classification systems [102]. It was initially annotated as a precursor miRNA (pre-mir-886) based on sequencing detection of putative small RNAs miR-886-5p and miR-886-3p; subsequent studies reclassified nc886 as a vtRNA-like ncRNA [17]. However, substantial experimental evidence now demonstrates that nc886 neither generates functional mature miRNAs nor serves as an integral component of vault complexes [103]. Collectively, these findings strongly support reclassifying nc886 as a distinct subclass of Pol III-transcribed genes.

RNA polymerase III-transcribed non-coding RNAs perform indispensable functions in a wide array of cellular processes, including translation, RNA maturation, epigenetic regulation, and stress response. These roles are mediated through tightly regulated transcriptional mechanisms and distinct secondary structural features.

## 3. Diseases Associated with Pol III Transcription

Dysregulation of Pol III-transcribed RNA—whether through aberrant transcription, faulty post-transcriptional modifications, or impaired processing—has been mechanistically implicated in various human diseases, particularly cancer (Figure 3). Recent studies investigating the roles of Pol III-derived RNAs in disease contexts are summarized in Table 1. In the following sections, we systematically discuss the role of Pol III-transcribed RNAs in disease pathogenesis.

### 3.1. Cancer

Most Pol III transcription is frequently upregulated in cancer cells, where it supports the increased demand for protein synthesis and ribosome biogenesis. Overexpression of Pol III-transcribed RNAs (e.g., tRNA, 5S RNA, U6, Y RNA, vtRNA, 7SL, Alu, BC200, snaR, RPPH1, and RMRP) has been reported across multiple malignancies such as breast, lung, and colorectal cancers [17]. In contrast, decreased levels of certain Pol III-transcribed RNAs, such as 7SK, have also been documented, further underscoring the complex role of Pol III transcriptional dysregulation in tumorigenesis [27].

Pol III-transcribed ncRNAs exhibit functional diversity and mechanistic specificity that contribute to cancer pathogenesis. Subtle alterations in tRNA pools profoundly impact cellular physiology and contribute to diverse human pathologies [43]. Perturbations in tRNA libraries arise from multiple mechanisms, including mutations in tRNA genes, transcriptional defects, maturation errors, and dysregulated post-transcriptional modifications [20,104]. For example, destabilization of N^7^-methylguanosine (m^7^G)-modified tRNAs, a conserved epitranscriptomic mark, reduces tRNA stability and abundance [46]. Notably, METTL1-mediated m^7^G modification drives leukemogenesis in acute myeloid leukemia (AML) by reprogramming tRNA-dependent translational control, illustrating how epitranscriptomic dysregulation interfaces with oncogenic pathways [105]. Similarly, METTL1/WDR4-mediated tRNA m^7^G modification enhances osteosarcoma progression and chemoresistance by altering oncogenic mRNA translation [106], while METTL1 depletion reduces m^7^G-modified tRNA levels in esophageal squamous cell carcinoma, impairing translation of oncogenic transcripts in the RPTOR/ULK1/autophagy axis [107]. Furthermore, tRNA modification changes can generate functional tRNA fragments. METTL1-mediated m^7^G hypomethylation promotes tsRNA biogenesis in prostate cancer [108], and m^7^G-modified tsRNA-LysTTT catalyzed by METTL1 enhances bladder cancer malignancy [109]. tsRNAs have been implicated in both tumor-promoting and tumor-suppressive roles. For instance, HCETSR (tRNA-Glu/TTC-derived) suppresses hepatocellular carcinoma via the SPTBN1/catenin axis [110], whereas tRF-23-Q99P9P9NDD promotes gastric cancer progression by modulating lipid metabolism and ferroptosis [111]. Conversely, tRF-33-P4R8YP9LON4VDP inhibits gastric cancer by regulating STAT3 signaling in an AGO2-dependent manner [112]. The 5′tRNA derivative tRF-Tyr competitively binds hnRNPD to modulate the c-Myc/Bcl2/Bax pathway, suppressing gastric cancer [113]. In breast cancer, 3′tRF-AlaAGC activates NF-κB signaling via TRADD interaction to drive malignancy and macrophage M2 polarization [114], while 3′-pre-tRNA-derived tRF-1-Ser enhances proliferation and stemness by inhibiting MBNL1 [115]. Additionally, tsRNA-GlyGCC promotes colorectal cancer progression and 5-FU resistance through SPIB regulation [116]. These findings underscore the multifaceted roles of tRNAs and tsRNAs in cancer.

Beyond tRNAs and tsRNAs, other Pol III transcripts are implicated in tumorigenesis. VtRNA, particularly vtRNA1-1, promotes liver cancer progression by interacting with TRIM21 [117]. RPPH1 enhances breast cancer progression by stabilizing m^6^A-modified FGFR2 mRNA via IGF2BP2, activating PI3K/AKT signaling [87]. RMRP accelerates esophageal squamous cell carcinoma through the miR-580-3p/ATP13A3 axis [118] and promotes ovarian cancer invasion via RAB31-dependent MMP secretion [119]. The m^6^A-modified 7SK snRNA regulates Pol II transcription in non-small cell lung cancer through P-TEFb [120], with 7SK overexpression suppressing tumor migration and invasion [121]. Alu RNA induces epithelial-to-mesenchymal transition in colorectal cancer via NLRP3 inflammasome activation and IL-1β release [122], while BC200 promotes EBV-associated nasopharyngeal carcinoma by sequestering miR-6834-5p to upregulate thymidylate synthase [123]. Of particular interest is nc886, a unique Pol III transcript with context-dependent roles in cancer. Typically silenced by CpG promoter hypermethylation in various malignancies, nc886 acts as a tumor suppressor by modulating immune responses in prostate cancer [124,125]. However, in ovarian cancer, TGF-β-induced promoter demethylation reactivates nc886 expression, promoting aggressive tumor progression [126]. This dual role highlights the epigenetic plasticity and regulatory complexity of nc886 in cancer biology.

### 3.2. Viral Infection

Emerging evidence underscores the multifaceted roles of RNA Pol III-transcribed RNAs in viral pathogenesis, where these non-coding RNAs are frequently hijacked by viruses to manipulate host cellular pathways or evade immune surveillance. For instance, during HIV infection, U6 snRNA translocates from the nucleus into extracellular vesicles, facilitating viral dissemination through intercellular communication [127]. Concurrently, the ADF-1L protein, derived from the PIF/pioneer transposon, upregulates 7SL RNA expression to bolster host innate immunity against pathogens, illustrating a counteractive host defense mechanism [128]. During Epstein–Barr virus (EBV) infection, BC200 RNA stabilizes EIF4A3 to modulate viral and host gene expression, suggesting a dual role in maintaining viral latency and cellular homeostasis [129]. Conversely, during Kaposi’s sarcoma-associated herpesvirus (KSHV) infection, the DUSP11-regulated nc886 represses interferon-stimulated genes (ISGs) to suppress antiviral responses, thereby creating a permissive environment for viral replication [130]. These examples collectively reveal a dynamic interplay between viral strategies and host Pol III RNAs, where viruses either exploit these RNAs to enhance infectivity or are constrained by their immunomodulatory functions. Future studies should prioritize mapping tissue-specific Pol III RNA—virus interactomes and deciphering how epitranscriptomic modifications influence these interactions. Such efforts could unveil novel therapeutic targets, such as inhibitors of nc886-mediated ISG suppression or enhancers of 7SL RNA’s antiviral activity, to disrupt viral life cycles while preserving host defense integrity. Ultimately, a deeper understanding of the dual roles of Pol III-transcribed RNAs in viral infection may provide a foundation for RNA-based antiviral diagnostics and precision therapeutics.

### 3.3. Other Diseases

Emerging evidence highlights the expanding roles of RNA Pol III-transcribed RNAs in diverse non-cancer pathologies, spanning neurodegenerative, cardiovascular, autoimmune, and developmental disorders. These RNAs orchestrate disease mechanisms through epitranscriptomic regulation, RNA–protein interactions, and pathway modulation, revealing their systemic impact beyond oncogenesis.

In neurodegenerative disorders, Alzheimer’s disease (AD) exhibits amyloid pathology-driven disruption of protein homeostasis, linked to reduced ELP3 expression and impaired tRNA modification [131]. Complementing this, tRNA methyltransferase TRMT10A catalyzes N^1^-methylguanosine (m^1^G) at position 9 of tRNAs, with its deletion in mice impairing brain function and underscoring tRNA modification’s role in neurodevelopment [132]. Recent studies have identified important roles for tsRNAs in mRNA silencing, translational regulation, apoptosis inhibition, intercellular communication, and epigenetic regulation [133]. For example, AS-tDR-013428 is involved in AD patients by targeting *RPSA* mRNA through a miRNA-like pattern [134]. A recent study found that tRF-Ala-AGC-3-M8 binds to the *EPHA7* 3′ UTR region and inhibits *EPHA7* translation to attenuate neuroinflammation and neuronal damage in AD patients [135]. Beyond neurodegenerative conditions, cardiovascular pathologies involve context-specific tsRNAs: rno-tsr007330 modulates myocardial fibrosis via NAT10-mediated EGR3 mRNA acetylation [136], while tRNA-Cys-5-0007 reduces ocular angiogenesis and inflammation by targeting the 3′-UTR region of VEGFA and TGF-β1 to inhibit expression [137]. Conversely, tRF-Glu-CTC exacerbates neointimal hyperplasia after vascular injury and drives neovascular age-related macular degeneration [138,139], illustrating the dual regulatory potential of tsRNAs in vascular remodeling. In autoimmune diseases, tRF-His-GTG-1 enhances neutrophil extracellular trap formation and interferon-alpha production via extracellular vesicles in systemic lupus erythematosus [140], whereas tsRNA-Gln-i-0095 suppresses neuroinflammation by silencing NFIA and TGFBR2 via a miRNA-like mechanism [141]. Reproductive health also relies on Pol III RNA regulation: angiogenin-mediated tRNA cleavage generates stress-responsive tsRNAs, while epididymal RNase homologs (RNase 9-12) maintain murine fertility through controlled tRNA processing under physiological conditions [142].

Splicing fidelity across tissues depends critically on Pol III transcripts. In Alazami syndrome, reduced LARP7-mediated 2′-O-methylation of U6 snRNA disrupts alternative splicing [56], whereas in mice, this modification ensures spermatogenesis [57]. Similarly, THUMPD2-catalyzed N^2^-methylation of U6 snRNA regulates retinal integrity through pre-mRNA splicing control [143].

Stress adaptation mechanisms further showcase Pol III RNA versatility. While 7SL traditionally facilitates ER-directed protein translocation via signal recognition particles [144], acute heat shock induces 7SL to globally suppress transcription and translation independently of secretory pathways [145]. In respiratory diseases, BC200 overexpression in asthma patients mediates inflammatory responses, linking Pol III RNAs to airway pathology [146]. In addition, recent studies have found that 5S rRNA pseudogene transcripts are associated with interferon production and inflammatory responses in alcohol-associated hepatitis [147]. Skeletal disorders like cartilage–hair hypoplasia (CHH) arise from pathogenic RMRP variants, with recent studies elucidating its role in osteoarthritis via FOXC1-RBP4-JNK axis activation [148,149,150,151,152,153]. Additionally, RMRP drives ligamentum flavum hypertrophy by regulating Gli1 SUMOylation and GSDMD-mediated pyroptosis, expanding its pathogenic scope in skeletal remodeling [154].

Collectively, from neurodegeneration to autoimmune dysregulation, Pol III-transcribed RNAs emerge as multifaceted regulators of human disease. Their context-dependent roles, spanning epitranscriptomic modification, RNA splicing, and stress adaptation, highlight both therapeutic challenges and opportunities. Future studies dissecting tissue-specific RNA interactomes will be critical for translating these insights into precision medicine strategies.

**Table 1 ijms-26-05852-t001:** Regulatory mechanisms of the Pol III transcriptome in disease.

Identity of Pol III Transcriptome	Biological Mechanism	Disease Type	Ref.
tRNA	METTL1-mediated tRNA m^7^G modification promotes leukaemogenesis of AML via tRNA regulated translational control.	Acute myeloid leukemia	[105]
tRNA	METTL1/WDR4-mediated tRNA m^7^G modification and mRNA translation control promote oncogenesis.	Oncogenesis	[106]
tRNA	Amyloid pathology reduces ELP3 expression and tRNA modifications leading to impaired proteostasis.	Alzheimer’s disease	[131]
tRNA	tRNA methyltransferase TRMT10A catalyzes N^1^-methylguanosine (m^1^G) at position 9 of tRNAs, with its deletion in mice impairing brain function and underscoring tRNA modification’s role in neurodevelopment.	Brain dysfunction	[132]
tsRNA	m^7^G-modified tsRNA-LysTTT catalyzed by METTL1 enhances bladder cancer malignancy.	Bladder cancer malignancy	[109]
tsRNA	tRF-23-Q99P9P9NDD promotes gastric cancer progression by modulating lipid metabolism and ferroptosis.	Gastric cancer	[111]
tsRNA	HCETSR (tRNA-Glu/TTC-derived) suppresses hepatocellular carcinoma via the SPTBN1/catenin axis.	Hepatocellular carcinoma	[110]
tsRNA	tRF-33-P4R8YP9LON4VDP inhibits gastric cancer by regulating STAT3 signaling in an AGO2-dependent manner.	Gastric cancer	[112]
tsRNA	5′tRNA derivative tRF-Tyr competitively binds hnRNPD to modulate the c-Myc/Bcl2/Bax pathway, suppressing gastric cancer.	Gastric cancer	[113]
tsRNA	3′tRF-AlaAGC activates NF-κB signaling via TRADD interaction to drive malignancy and macrophage M2 polarization.	Breast cancer	[114]
tsRNA	3′-pre-tRNA-derived tRF-1-Ser enhances proliferation and stemness by inhibiting MBNL1.	Breast cancer	[115]
tsRNA	tsRNA-GlyGCC promotes colorectal cancer progression and 5-FU resistance through SPIB regulation.	Colorectal cancer	[116]
tsRNA	tRF-Ala-AGC-3-M8 binds to the *EPHA7* 3′ UTR region and inhibits *EPHA7* translation to attenuate neuroinflammation and neuronal damage in AD patients	Alzheimer’s disease	[135]
tsRNA	tsRNAs (rno-tsr007330) modulates myocardial fibrosis via NAT10-mediated EGR3 mRNA acetylation.	Neurodegenerative conditions	[136]
tsRNA	tRNA-Cys-5-0007 attenuates ocular angiogenesis and inflammation by targeting VEGFA and TGF-β1.	Neurodegenerative conditions	[137]
tsRNA	tRF-Glu-CTC exacerbates neointimal hyperplasia after vascular injury and drives neovascular age-related macular degeneration.	Macular degeneration	[138,139]
tsRNA	tRF-His-GTG-1 enhances neutrophil extracellular trap formation and interferon-alpha production via extracellular vesicles in systemic lupus erythematosus.	Systemic lupus erythematosus	[140]
tsRNA	tsRNA-Gln-i-0095 suppresses neuroinflammation by downregulating NFIA and TGFBR2 through miRNA-like mechanisms.	Neuroinflammation	[141]
tsRNA	Angiogenin-mediated tRNA cleavage generates stress-responsive tsRNAs, while epididymal RNase homologs (RNase 9-12) maintain murine fertility through controlled tRNA processing under physiological conditions.	Infertility	[142]
vtRNA	TRIM21 modulates stability of pro-survival, non-coding RNA vtRNA1-1 in human hepatocellular carcinoma cells.	Hepatocellular carcinoma	[117]
RPPH1	RPPH1 enhances breast cancer progression by stabilizing m^6^A-modified FGFR2 mRNA via IGF2BP2, activating PI3K/AKT signaling.	Breast cancer	[87]
RMRP	RMRP accelerates C through the miR-580-3p/ATP13A3 axis.	Esophageal squamous cell carcinoma	[118]
RMRP	RMRP promotes ovarian cancer invasion via RAB31-dependent MMP secretion.	Ovarian cancer	[119]
RMRP	LncRNA RMRP promotes chondrocyte injury by regulating the FOXC1/RBP4 axis.	Cartilage-hair hypoplasia syndrome	[148]
RMRP	RMRP variants inhibit the cell cycle checkpoints pathway in cartilage–hair hypoplasia.	Cartilage-hair hypoplasia syndrome	[149]
RMRP	The RMRP gene n.197C>T mutation causes cartilage–hair hypoplasia syndrome.	Cartilage-hair hypoplasia syndrome	[151]
RMRP	RMRP accelerates ligamentum flavum hypertrophy by regulating GSDMD-mediated pyroptosis through Gli1 SUMOylation.	Hypertrophy of ligamentum flavum	[154]
7SK snRNA	The m^6^A-modified 7SK snRNA regulates Pol II transcription in non-small cell lung cancer through P-TEFb.	Lung cancer	[120]
Alu RNA	Alu RNA induces epithelial-to-mesenchymal transition in colorectal cancer via NLRP3 inflammasome activation and IL-1β release.	Colorectal cancer	[122]
BC200	BC200 promotes EBV-associated nasopharyngeal carcinoma by sequestering miR-6834-5p to upregulate thymidylate synthase.	EBV-associated nasopharyngeal carcinoma	[123]
BC200	BC200 RNA stabilizes EIF4A3 to modulate viral and host gene expression, suggesting a dual role in maintaining viral latency and cellular homeostasis.	Epstein–Barr virus infection	[129]
BC200	BC200 overexpression in asthma patients mediates inflammatory responses, linking Pol III RNAs to airway pathology.	Asthma	[146]
nc886	nc886 acts as a tumor suppressor by modulating immune responses in prostate cancer.	Prostate cancer	[125]
nc886	TGF-β-induced CpG demethylation reactivates nc886 to drive aggressive ovarian cancer progression.	Ovarian cancer	[126]
nc886	The DUSP11-regulated nc886 represses interferon-stimulated genes to suppress antiviral responses, thereby creating a permissive environment for viral replication.	Kaposi’s sarcoma-associated herpesvirus infection	[130]
U6 snRNA	U6 snRNA translocates from the nucleus into extracellular vesicles, facilitating viral dissemination through intercellular communication.	HIV infection	[127]
U6 snRNA	THUMPD2-catalyzed N^2^-methylation of U6 snRNA regulates retinal integrity through pre-mRNA splicing control.	Age-related macular degeneration	[143]
7SL RNA	The ADF-1L protein, derived from the PIF/pioneer transposon, upregulates 7SL RNA expression to bolster host innate immunity against pathogens, illustrating a counteractive host defense mechanism.	Virus infection	[128]
7SL RNA	7SL RNA and signal recognition particle orchestrate a global cellular response to acute thermal stress.	Acute thermal stress	[145]
5S rRNA	5S rRNA pseudogene transcripts are associated with interferon production and inflammatory responses in alcohol-associated hepatitis.	Alcohol-associated hepatitis	[147]

## 4. Diagnostic and Therapeutic Strategies for Pol III Transcription-Related Diseases

The dual roles of Pol III-transcribed RNAs—as both drivers of pathogenesis and mediators of cellular homeostasis—have positioned them as promising targets for therapeutic innovation and diagnostic advancement. Emerging evidence underscores their potential as therapeutic targets through direct transcriptional inhibition or modulation of downstream pathways, while their remarkable stability in biofluids offers unprecedented opportunities for non-invasive disease detection. Below, we delineate the current landscape of Pol III-directed therapeutic strategies and biomarker development, highlighting their transformative potential in precision medicine.

### 4.1. Therapeutic Interventions

Therapeutic strategies targeting Pol III transcription are gaining momentum, driven by its central role in disease mechanisms. The potential of tRNAs as therapeutic agents has only recently been recognized, and no clinical studies have been registered to date [155]. However, two recent studies of sup-tRNA delivery in mice used delivery platforms developed for other RNAs, supporting their potential applicability to tRNA therapy [156,157]. tRNA therapeutics have the potential to treat a wide range of diseases, and although much of the current work is still in preclinical development, they are advancing toward clinical application [158], with several biotech companies actively exploring their therapeutic potential [159,160]. Beyond tRNAs, RMRP attenuates microglial apoptosis and promotes motor recovery after spinal cord injury via the EIF4A3/SIRT1 axis, highlighting its neuroprotective potential [161]. Similarly, adenoviral delivery of nc886, an anti-apoptotic Pol III transcript, enhances gene therapy efficacy by suppressing interferon responses [162]. Such strategies underscore the potential of Pol III RNAs as therapeutic targets and agents across a wide array of pathological contexts, including neurodegeneration, inflammation, and immune dysregulation. Further work is needed to refine delivery systems, ensure tissue specificity, and mitigate off-target effects to fully unlock the therapeutic promise of these transcripts.

### 4.2. Diagnostic Biomarkers

Pol III-transcribed RNAs display remarkable stability in biological fluids such as blood, plasma, urine, and extracellular vesicles, making them powerful non-invasive biomarkers. Several classes of these RNAs (including tsRNAs, RMRP, Y RNA, and vtRNA) have shown significant potential as diagnostic biomarkers across a spectrum of diseases (Table 2). In cancer diagnostics, tsRNAs have been demonstrated as promising biomarkers in gastric cancer, breast cancer, lung cancer, and colorectal cancer [163]. For instance, in gastric cancer, multiple tsRNAs exhibit differential expression patterns: tRF-24-6VR8K09LE9 and i-tRF-AsnGTT are downregulated in serum [164,165], while tRF-31-PNR8YP9LON4VD, tRF-30-MIF91SS2P4FI, tRF-30-IK9NJ4S2I7L7, tRF-17-18VBY9M, and has-tsr013526 are upregulated in serum or tissues [166,167,168]. Similarly, in hepatocellular carcinoma, upregulated tsRNAs such as tRF-23-R9J89O9N9, tRF-33-RZYQHQ9M739P0J, tsRNA-Thr-5-0015, and tRF-3a-Pro are detected in serum or tissues [169,170,171,172]. Beyond cancer, tiRNA-Gly-GCC-001 (serum, upregulated) is linked to major depressive disorder [173], and tiRNA-Gln-CTG (plasma, downregulated) is associated with pre-eclampsia [174]. Additionally, under non-cancer conditions like nonproliferative diabetic retinopathy, 5′tiRNA-35-PheGAA-8, tRF3-28-PheGAA-1, and tRF3b-PheGAA-6 are elevated in peripheral blood mononuclear cells [175]. These findings collectively highlight the diverse diagnostic potential of tsRNAs across both oncological and non-oncological diseases, underscoring their promise as biomarkers in multiple clinical contexts.

RMRP RNA demonstrates clinical relevance in cancer, cardiovascular, and neuropsychiatric disorders. For instance, exosomal RPPH1 and RMRP in serum correlate with immune infiltration in gastric cancer [176], while urinary exosomal RMRP detected via RT-RAA-CRISPR/Cas12a enables non-invasive bladder cancer screening [177]. Elevated RMRP levels in serum are further associated with coronary artery disease [178] and acute coronary syndrome [179], and its upregulation in peripheral blood mononuclear cells correlates with bipolar disorder [180]. Beyond RMRP, other Pol III RNAs show diverse applications. VtRNA1-1 levels in serum reflect bone marrow activity, serving as a hematologic malignancy biomarker [181], and serum exosomal BC200 levels decline post-resection in bladder cancer patients, suggesting utility for monitoring therapeutic response [182]. Notably, reduced leukocyte RPPH1 levels predict pre-eclampsia risk [183], and exosomal Alu RNA promotes colorectal tumorigenesis while serving as a diagnostic biomarker, exemplifying theranostic potential [122]. In neurological disorders, elevated serum BC200 and SNHG3 levels distinguish multiple sclerosis patients, linking Pol III dysregulation to neuroinflammation [184]. Despite these promising findings, several challenges remain—most notably the standardization of detection methodologies and the characterization of tissue-specific expression profiles. Large-scale validation studies and mechanistic investigations will be crucial for clinical translation.

In conclusion, the Pol III transcriptome holds vast diagnostic and therapeutic potential across a diverse array of diseases. From RNA-targeted therapies to biomarker-guided diagnosis, advances in understanding and manipulating Pol III-derived RNAs are paving the way toward RNA-centric strategies in precision medicine. Continued integration of mechanistic research with clinical validation will be essential to realize the full translational impact of these non-coding RNA species.

**Table 2 ijms-26-05852-t002:** Pol III-transcribed RNAs as diagnostic biomarkers for various diseases.

Identity of Pol III Transcriptome	Type	Source	Expression Level	Diagnostic Type of Disease	Ref.
tsRNA	tsRNA-Gly-CCC-2, tsRNA-Gly-GCC-1, and tsRNA-Lys-CTT-2-M2	serum	up	tuberculosis	[185]
tsRNA	tRF-22-RNLNK88KL, tRF-27-Z3M8ZLSSXUL, and tRF-32-0668K87SERM4P	tissues and plasma	up	colorectal cancer	[186]
tsRNA	tRF-24-6VR8K09LE9	serum	down	gastric cancer	[164]
tsRNA	tRF-31-PNR8YP9LON4VD, tRF-30-MIF91SS2P4FI, and tRF-30-IK9NJ4S2I7L7	serum	up	gastric cancer	[166]
tsRNA	tRF-17-18VBY9M	tissues and serum	up	gastric cancer	[167]
tsRNA	has-tsr013526	serum	up	gastric cancer	[168]
tsRNA	tiRNA-Gly-GCC-001	serum	up	major depressive disorder	[173]
tsRNA	has-tsr011468	tissues and serum	Up	lung adenocarcinoma	[187]
tsRNA	5′-tRNA-Glu-TTC-9-1_L30, 5′-tRNA-Val-CAC-3-1_L30, and 5′-M-tRNA-Gln-TTG-3-3_L30	serum and semen	up	prostate cancer	[188]
tsRNA	tRF-1:28-chrM.Ser-TGA and tiRNA-1:34-Glu-CTC-1-M2	plasma	up	bladder cancer	[189]
tsRNA	tRF-23-R9J89O9N9	serum	up	hepatocellular carcinoma	[169]
tsRNA	tiRNA-Gln-CTG	plasma	down	pre-eclampsia	[174]
tsRNA	tRF-33-RZYQHQ9M739P0J	tissues and serum	up	hepatocellular carcinoma	[170]
tsRNA	tsRNA-Thr-5-0015	serum	up	hepatocellular carcinoma	[171]
tsRNA	tRF-3a-Pro	serum	up	hepatocellular carcinoma	[172]
tsRNA	5′tiRNA-35-PheGAA-8, tRF3-28-PheGAA-1, tRF3b-PheGAA-6, mt-tRF3-19-ArgTCG, mt-tRF3-20-ArgTCG, and mt-tRF3-21-ArgTCG	peripheral blood mononuclear cells	up (5′tiRNA-35-PheGAA-8, tRF3-28-PheGAA-1, and tRF3b-PheGAA-6)/down (mt-tRF3-19-ArgTCG, mt-tRF3-20-ArgTCG, and mt-tRF3-21-ArgTCG)	nonproliferative diabetic retinopathy	[175]
tsRNA	tsRNA-49-73-Glu-CTC	serum	up	non-small cell lung cancer	[190]
tsRNA	i-tRF-AsnGTT	serum	down	gastric cancer	[165]
vtRNA	vtRNA1-1	serum	up	hematological disorders	[181]
RPPH1 and RMRP RNAs	RPPH1 and RMRP	serum	up	gastric cancer	[176]
RPPH1 RNA	RPPH1	plasma/leukocytes	up (plasma)/down (leukocytes)	pre-eclampsia	[183]
RMRP RNA	RMRP	urine	up	bladder cancer	[177]
RMRP RNA	RMRP	serum	up	coronary artery disease	[178]
RMRP RNA	RMRP	peripheral blood mononuclear cells	up	bipolar disorder	[180]
RMRP RNA	RMRP	serum	up	acute coronary syndrome	[179]
BC200	BCYRN1	serum	up	bladder cancer	[182]
BC200	BCYRN1	serum	up	multiple sclerosis	[184]
Alu RNA	Alu	exosomes in serum	up	colorectal cancer	[122]
Y RNA	Y RNA	plasma	up	colorectal cancer	[191]

## 5. Conclusions and Further Perspectives

The expanding repertoire of Pol III-transcribed ncRNAs underscores their pivotal, yet intricate, roles in human health and disease. These RNAs orchestrate fundamental cellular processes such as translation, splicing, stress adaptation, and epigenetic regulation. Recent discoveries have illuminated the multifaceted nature of these RNAs, revealing their context-dependent behavior that can either maintain cellular homeostasis or drive pathogenic transformation under diseased conditions. Dysregulation of Pol III transcription or RNA processing has been implicated in a wide spectrum of diseases, including cancer, neurodegenerative disorders, autoimmune diseases, and viral infections. These findings reframe Pol III transcripts as not merely structural components, but as central regulators of cellular and pathological states, offering new insights into disease mechanisms and identifying novel molecular entry points for diagnosis and intervention.

From a diagnostic perspective, the stability of Pol III-derived RNAs in biofluids, such as serum tsRNAs and exosomal Alu RNA, offers promising avenues for non-invasive disease detection, exemplified by tuberculosis-specific tsRNA signatures and bladder cancer-associated BC200 levels. On the therapeutic front, emerging strategies targeting the Pol III transcriptome, such as tRNA therapeutics, have the potential to treat a wide range of diseases, and although much of the current work is still in preclinical development, they are moving closer to the clinical phase [158]. However, challenges remain in addressing tissue-specific RNA isoform redundancy, optimizing delivery systems, and standardizing biomarker assays for clinical translation.

Looking ahead, several key directions are poised to shape the next phase of Pol III transcriptome research. A critical priority is the mechanistic resolution of how Pol III-transcribed RNAs function within dynamic cellular contexts, particularly through the lens of epitranscriptomic modifications such as m^7^G and m^1^G, which may fine-tune RNA stability, localization, and interaction networks in disease-specific settings. Therapeutic innovation will depend on the development of delivery systems capable of targeting specific tissues or cell types, as well as the integration of RNA-based interventions with existing therapeutic modalities, such as immune checkpoint inhibitors or epigenetic drugs. On the diagnostic front, efforts should focus on validating multi-RNA biomarker panels across large, diverse clinical cohorts and refining detection technologies—such as CRISPR–Cas platforms—to achieve higher sensitivity and specificity in clinical settings. Understanding the interface between viral pathogenesis and Pol III RNA regulation may also unveil novel vulnerabilities for antiviral intervention, while comparative studies across species can provide valuable evolutionary insights into the conserved roles of these RNAs in cellular regulation and disease progression. Collectively, these future directions aim to bridge basic mechanistic discoveries with translational applications, ultimately establishing the Pol III transcriptome as a central axis in RNA-based precision medicine.

## Figures and Tables

**Figure 1 ijms-26-05852-f001:**
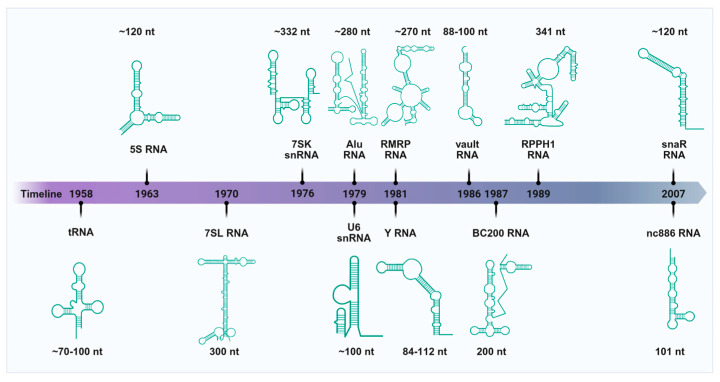
Historical timeline of studies related to the identification of the Pol III transcriptome.

**Figure 2 ijms-26-05852-f002:**
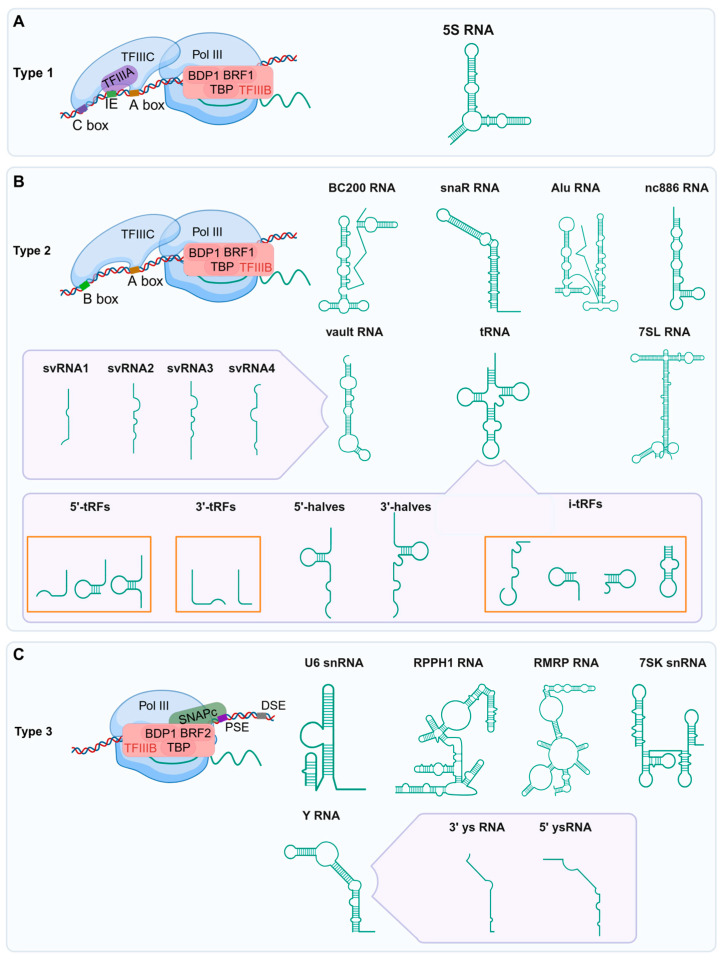
Different promoter types for Pol III transcriptome-dependent gene transcription. (**A**) The type 1 promoter contains an internal control region downstream of the transcription start site, where TFIIIA binds to, recruits TFIIIC and TFIIIB, and ultimately localizes Pol III. The type 1 promoter is well conserved from yeast to humans and is restricted to the 5S rRNA gene. (**B**) Type 2 promoters similarly utilize internal control regions but bypass TFIIIA, relying solely on TFIIIC for TFIIIB recruitment. tRNA, vtRNA, BC200, snaR, Alu, nc886, and 7SL RNA are all transcribed from type 2 promoters. In addition, vault RNAs derive fragments that are svRNA1, svRNA2, svRNA3, and svRNA4. Small fragments of tRNA-derived tsRNAs were categorized into five classes: 5′-tRFs, 3′-tRFs, 5′-halves, 3′-halves, and i′-tRFs. (**C**) Type 3 promoters are found in higher eukaryotes (not in yeast) and consist of a proximal sequence element (PSE), a TATA box, and a distal sequence element (DSE). Type 3 promoters consist of elements that direct the assembly of transcription complexes entirely upstream of the TSS. U6 snRNA, 7SKsnRNA, RPPH1, RMRP, and Y RNA are all transcribed from type 3 promoters. In particular, Y RNAs derive fragments that are 3′ysRNAs and 5′ysRNAs. TFIIIB is a complex formed by BRF1 (or BRF2), TBP (TATA-binding protein), and BDP1.

**Figure 3 ijms-26-05852-f003:**
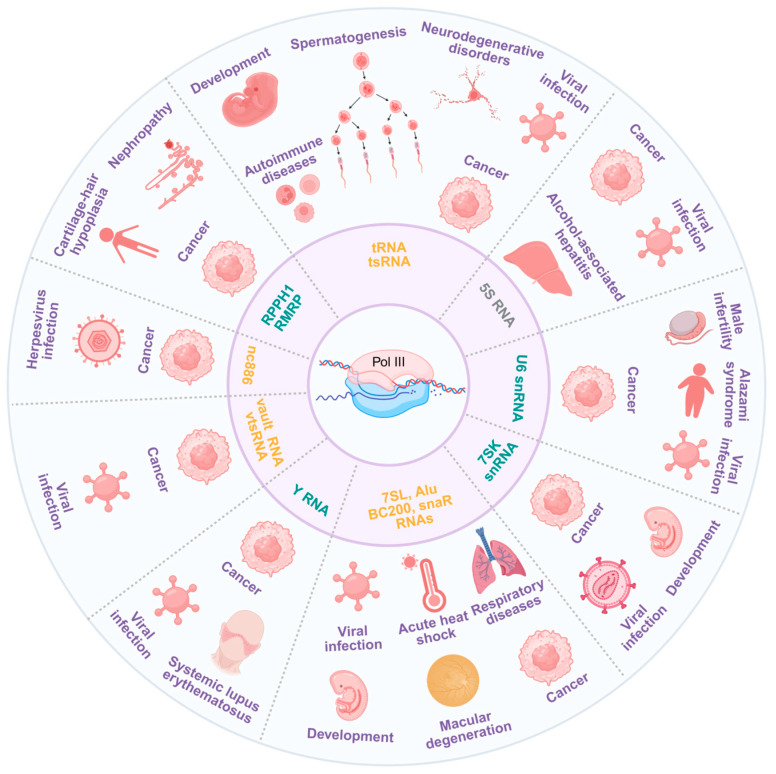
Pol III-transcribed RNAs are associated with a multitude of diseases.

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
