# Peer review of "RNA Polymerase III-Transcribed RNAs in Health and Disease: Mechanisms, Dysfunction, and Future Directions"

_ijms, 2025, doi:10.3390/ijms26125852_

Round 1
Reviewer 1 Report
Comments and Suggestions for Authors
In this manuscript, Sun et al. present a comprehensive and well-organised review of the RNA polymerase III (Pol III) transcriptome. The review provides an overview of the biogenesis, regulatory mechanisms and disease associations of Pol III-transcribed RNAs. It is clearly written, well-referenced and covers both very recent and older, but relevant literature.
I only have a few minor comments that I would like the authors to address prior to publication:
- Line 96: The statement “vtRNA can be derived from 3’ysRNA and 5’ysRNA” appears to be incorrect. If this is an error, it should be removed. If indeed this can be supported, please provide clarification in the text and cite appropriate supporting references.
- Section 2.6 - RMRP RNA: RNAse MRP also has a functional role in mitochondria. This mitochondrial function should also be briefly mentioned, along with the appropriate reference.
- Section 3.1 – Cancer: Although the authors mention “Subtle alterations in tRNA pools”, this section would benefit from a brief discussion of the work by Tavazoie SF and colleagues (also see Pinzaru & Tavazoie SF, Nat Rev Cancer, 2023) on the overexpression of specific tRNAs (or tRNA pools) and tRNA stability on cancer progression and metastasis. Additionally, including a brief discussion of the work by White RJ and colleagues (see White RJ, Nat Rev Mol Cell Biol, 2005) on the regulation of Pol III transcription by tumour suppressors and oncogenes would provide valuable mechanistic context.
- Line 491 – typo: “has-tsr013526” should be “hsa- tsr013526”.
Author Response
Response to Reviewer 1 Comments
Comments 1: Line 96: The statement “vtRNA can be derived from 3’ysRNA and 5’ysRNA” appears to be incorrect. If this is an error, it should be removed. If indeed this can be supported, please provide clarification in the text and cite appropriate supporting references.
Response 1: Thank you for the helpful comments. Here, the fragments produced by vtRNA include svRNA1, svRNA2, svRNA3, and svRNA4, and the fragments produced by Y RNA include 3'ysRNA and 5'ysRNA. therefore, we have corrected the figure legends in the revised version (Page 4, Line 101-107).
Comments 2: Section 2.6 - RMRP RNA: RNAse MRP also has a functional role in mitochondria. This mitochondrial function should also be briefly mentioned, along with the appropriate reference.
Response 2: Thank you for pointing this out. We agree with this comment. Therefore, we have described the function and mechanism of RMRP in mitochondria and provide the corresponding references in revised version (Page 6 and 7, Line 247-250).
Comments 3: Section 3.1 – Cancer: Although the authors mention “Subtle alterations in tRNA pools”, this section would benefit from a brief discussion of the work by Tavazoie SF and colleagues (also see Pinzaru & Tavazoie SF, Nat Rev Cancer, 2023) on the overexpression of specific tRNAs (or tRNA pools) and tRNA stability on cancer progression and metastasis. Additionally, including a brief discussion of the work by White RJ and colleagues (see White RJ, Nat Rev Mol Cell Biol, 2005) on the regulation of Pol III transcription by tumour suppressors and oncogenes would provide valuable mechanistic context.
Response 3: We thank the reviewer for the professional suggestions. Following this advice, we have discussed and updated the text in revised version (Page 9, Line 338-342).
Comments 4: Line 491 – typo: “has-tsr013526” should be “hsa- tsr013526”.
Response 4: Thank you for the professional comments. Following this advice, we have corrected the text and table 2 in this revised version (Page 12, Line 504; Table 2).
Reviewer 2 Report
Comments and Suggestions for Authors
Sun et al. review PolII transcribed RNAs and provide an overview of their nature and cellular functions that are currently known. The text is well organized and quite comprehensive. I have only minor comments for potential improvements.
This review could be strengthened by including select examples of molecular mechanisms of regulation by PolIII RNAs. For example, in section 3.3, there are many studies cited for tsRNAs in disease. This is, however, just a list of molecules and diseases. Do all of these tsRNAs regulate their targets by a similar mechanism? How do they regulate in the first place? Are the molecular mechanisms known for all of the cited studies or are these just correlations? Providing one or a few specific examples of tsRNA function would be helpful.
Some of the more specific sub-classes of molecules are only indirectly introduced in the legend to Figure 2 but not in the text leading up to Fig. 2. These include 3’ysRNA, 5’ysRNA, 5’-tRF, 3’-tRF, i’-tRF, for example. Explicit introductions/definitions of these terms earlier in the text would make this better accessible to more general readers.
Figure 2: would be nice to label TFIIIB in addition to the labels for the individual components within TFIIIB in the three panels.
Author Response
Response to Reviewer 2 Comments
Comments 1: This review could be strengthened by including select examples of molecular mechanisms of regulation by PolIII RNAs. For example, in section 3.3, there are many studies cited for tsRNAs in disease. This is, however, just a list of molecules and diseases. Do all of these tsRNAs regulate their targets by a similar mechanism? How do they regulate in the first place? Are the molecular mechanisms known for all of the cited studies or are these just correlations? Providing one or a few specific examples of tsRNA function would be helpful.
Response 1: Thank you for the professional suggestions. Following this advice, we have included selected examples of molecular mechanisms regulated by PolIII RNAs, particularly tsRNAs. tsRNAs have been found to participate in biological processes by regulating target genes through a variety of pathways, including mRNA silencing, translational regulation, inhibition of apoptosis, intercellular communication, and epigenetic inheritance. We have added the most recent references and added them in Table 1. In addition, we have refined the specific target regulatory mechanisms of tsRNAs. All relevant content is incorporated in the revised version (page 10 and 11, lines 421-436; Table 1).
Comments 2: Some of the more specific sub-classes of molecules are only indirectly introduced in the legend to Figure 2 but not in the text leading up to Fig. 2. These include 3’ysRNA, 5’ysRNA, 5’-tRF, 3’-tRF, i’-tRF, for example. Explicit introductions/definitions of these terms earlier in the text would make this better accessible to more general readers.
Response 2: Thank you very much for the helpful comments and suggestions. Following this suggestion, we have included these small fragments (e.g. 3'ysRNA, 5'ysRNA, 5'-tRF, 3'-tRF, i'-tRF, etc.) in detail before Figure 2. Relevant text has been incorporated, described and highlighted (page 3, lines 85-92).
Comments 3: Figure 2: would be nice to label TFIIIB in addition to the labels for the individual components within TFIIIB in the three panels.
Response 3: Thank you for pointing this out. I/We agree with this comment. Therefore, we have labelled TFIIIB in addition to the individual components of TFIIIB in the three panels (Figure 2).